



# Response of O₂ and pH to ENSO in the California Current System in a high resolution global climate model

Giuliana Turi[1], Michael Alexander[2], Nicole S. Lovenduski[3], Antonietta Capotondi[2], James Scott[4], Charles Stock[5], John Dunne[5], Jasmin John[5], and Michael Jacox[6]

[1]Boulder, CO, USA
[2]NOAA/ESRL, Boulder, Colorado, USA
[3]Department of Atmospheric and Oceanic Sciences and Institute of Arctic and Alpine Research, University of Colorado, Boulder, Colorado, USA
[4]CIRES, University of Colorado at Boulder, and NOAA/ESRL, Boulder, Colorado, USA
[5]NOAA/GFDL, Princeton, New Jersey, USA
[6]University of California, Santa Cruz, CA and NOAA/SWFSC, Monterey, CA, USA

*Correspondence to:* Dr. Michael Alexander (michael.alexander@noaa.gov)

**Abstract.** We use a novel, high-resolution global climate model (GFDL-ESM2.6) to investigate the influence of warm and cold El Niño/Southern Oscillation (ENSO) events on the physics and biogeochemistry of the California Current System (CalCS). We focus on the effect of ENSO on variations in the O₂ concentration and the pH of the coastal waters of the CalCS. An assessment of the CalCS response to six El Niño and seven La Niña events in ESM2.6 reveals significant variations in the

response between events. However, these variations overlay a consistent physical and biogeochemical (O₂ and pH) response in the composite mean. Focusing on the mean response, our results demonstrate that O₂ and pH are affected rather differently in the euphotic zone above ∼100 m. The strongest O₂ response reaches up to several 100 km offshore, whereas the pH signal occurs only within a ∼100 km-wide band along the coast. By splitting the changes in O₂ and pH into individual physical and biogeochemical components that are affected by ENSO variability, we found that O₂ variability in the surface

ocean is primarily driven by changes in surface temperature that affect the O₂ solubility. In contrast, surface pH changes are predominantly driven by changes in dissolved inorganic carbon (DIC), which in turn is affected by upwelling, explaining the confined nature of the pH signal close to the coast. Below ∼100 m, we find conditions with anomalously low O₂ and pH, and by extension also anomalously low aragonite saturation, during La Niña. This result is consistent with findings from previous studies and highlights the stress that the CalCS ecosystem could periodically undergo in addition to impacts due to climate

change.

## 1   Introduction

Ocean deoxygenation (decreasing O₂ concentration) and ocean acidification (decreasing pH) are considered to be major oceanic ecosystem stressors that can severely reduce habitat suitability in benthic and pelagic ecosystems (e.g., Doney et al., 2009a; Doney, 2010; Gruber, 2011; Bopp et al., 2013; Breitburg et al., 2015). Coastal upwelling ecosystems, such as the

California Current System (CalCS), support some of the world's most productive fisheries due to the seasonal, wind-driven



upwelling of nutrient-rich waters (e.g., Chavez and Messié, 2009; Messié et al., 2009). The upwelled waters fuel biological production in the CalCS but are also characterized by lower $O_2$ and lower pH than the surrounding surface waters (e.g., Bograd et al., 2008; Nam et al., 2011). Although the CalCS is accustomed to seasonal fluctuations in $O_2$ and pH, several observational studies suggest that both have decreased in the last decades, particularly within 100 km of the coast (e.g., Bograd et al., 2008;

Feely et al., 2008; Nam et al., 2011; Bednaršek et al., 2014; Bednaršek and Ohman, 2015). Specifically, Bograd et al. (2008) found a shoaling of the hypoxic threshold, which is typically considered to be $\sim$60 μmol kg$^{-1}$ (Gray et al., 2002; Diaz and Rosenberg, 2008; Vaquer-Sunyer and Duarte, 2008), of up to 90 m in the southern CalCS between 1984 and 2006. Furthermore, Feely et al. (2008) report an incidence of nearshore surface waters becoming corrosive during the early upwelling season (May/June) of 2007, with the saturation state of aragonite ($\Omega_{arag}$) dropping below one, corresponding to pH < 7.75 (aragonite

is a mineral form of calcium carbonate, $CaCO_3$, found in the shells of certain calcifying marine organisms). Turi et al. (2016) used a regional ocean model to demonstrate that a recent intensification of upwelling-favorable winds is linked to a drop in coastal pH and $\Omega_{arag}$. It currently remains unclear whether these observed and modeled changes in the CalCS' ocean biogeochemistry are ongoing signals of anthropogenic climate change, and thus could continue into the future, or whether they are driven by natural fluctuations in the climate system (e.g., Bakun, 1990; Narayan et al., 2010; Bakun et al., 2015; Rykaczewski

et al., 2015; Wang et al., 2015).

The eastern North Pacific region is affected by several teleconnective climate patterns, including the El Niño Southern Oscillation (ENSO; e.g., Alexander et al., 2002; Mantua and Hare, 2002; Di Lorenzo et al., 2008; Alexander, 2010; Di Lorenzo et al., 2010; Macias et al., 2012; Franks et al., 2013; Capotondi et al., 2015; Newman et al., 2016). ENSO is a major driver of interannual physical variability and affects a range of variables on a local scale in the CalCS, such as sea-surface temper-

ature, thermocline depth, and the intensity and depth of upwelling (e.g., Chavez et al., 2002; Schwing et al., 2005; Checkley Jr. and Barth, 2009; Jacox et al., 2015b), and operates remotely through two distinct pathways: (i) through the atmosphere by affecting the intensity and location of the Aleutian Low pressure system and the fluxes of momentum, heat, and freshwater through the surface ocean and (ii) through the ocean by generating thermocline anomalies in the tropical Pacific that propagate eastward along the equator and northward as coastally trapped waves. During a typical ENSO warm event (El Niño), the

atmospheric influence is twofold: surface heat fluxes into the ocean are anomalously high and the Aleutian Low intensifies and is displaced southeastward, causing a decrease in equatorward, upwelling-favorable winds in the CalCS. From the oceanic side, the coastally trapped waves cause a depression of the thermo- and nutricline in the nearshore region, leading to reduced upwelling of nutrient-rich, cool waters and thus limiting biological production in the surface waters (Bograd and Lynn, 2001). Source waters for upwelling in the CalCS also tend to be anomalously warm, shallow, and fresh during El Niño (Jacox et al.,

2015b), resulting in similarly warm and fresh anomalies at the surface. During an ENSO cold event (La Niña), these processes are typically reversed, with nearshore surface waters being anomalously low in $O_2$ and pH due to a shoaling of the isopycnal surfaces (Nam et al., 2011).

A number of studies have looked at the influence of ENSO on the physics, biogeochemistry, and biology of the CalCS over the past several decades. For instance, the influence of the strong 1997/1998 El Niño and subsequent La Niña on the coastal

upwelling ecosystem of the US West Coast has been well documented by a variety of observational studies that focus on the



physics and hydrography (e.g., Collins et al., 2002; Kosro, 2002; Lynn and Bograd, 2002; Ryan and Noble, 2002; Schwing et al., 2002b), on nutrients and primary production (e.g., Castro et al., 2002; Kahru and Mitchell, 2000, 2002; Kudela and Chavez, 2002), and on zooplankton and higher trophic levels (e.g., Benson et al., 2002; Hopcroft et al., 2002; Pearcy, 2002; Peterson et al., 2002). A handful of observational studies have investigated the impact of the 1982/1983 El Niño on the physical

regime of the CalCS (e.g., Simpson, 1983; Huyer and Smith, 1985), and the effect of the 2002/2003 El Niño on the physics and biology (e.g., Murphree, 2003; Schwing et al., 2002a). Several regional modeling studies have shown the connection between ENSO and the vertical transport, water column density, origins and properties of upwelled water (Jacox et al., 2014, 2015b), and demonstrated the relative importance of remote versus local forcing on both the physics and the biogeochemistry of the CalCS (Frischknecht et al., 2015; Jacox et al., 2015a). During the 2010/2011 La Niña, Nam et al. (2011) observed decreases

in $O_2$ and pH in the upwelling region along the coast that were 2-3 times larger than expected solely due to the cross-shore shoaling of the isopycnal surfaces. They found that the additional reduction of $O_2$ was related to decreased subsurface primary production and a short-term strengthened poleward flow of the California Undercurrent. In addition, pH dropped below the critical value of 7.75 both during the upwelling season (typically April-September) and the two following La Niña months. These results suggest that severe low-$O_2$ and low-pH conditions may occur if La Niña conditions overlap with seasonal up-

welling.

To our knowledge, the study by Nam et al. (2011) is the only one to investigate the influence of ENSO on both $O_2$ and pH in the CalCS. It is only recently that CalCS ENSO responses have begun to be monitored by modern oceanographic and biogeochemical measurements. Moreover, we have yet to come across any studies investigating responses across multiple ENSO events. Earth System Models (ESMs), such as those participating in the Climate Model Intercomparison Project Phase 5 (CMIP5), pro-

vide an opportunity to address these limitations. Global ESMs used for multi-centennial climate change simulations, however, simulate the ocean at a fairly coarse horizontal resolution ($\sim$1° Stock et al., 2011). Models with such resolutions are challenged to reproduce the responses of coastal ecosystems to basin-scale ocean variability (Saba et al., 2016). It is commonly acknowledged that at least 0.1° horizontal ocean resolution is necessary to resolve the Rossby radius of deformation (e.g., Fiechter et al., 2014; Dunne et al., 2015), which is around 20-60 km in the CalCS (Chelton et al., 1998). In this study, we use a high-

resolution (0.1°), fully-coupled climate model developed at the Geophysical Fluid Dynamics Laboratory (GFDL-ESM2.6) to investigate the effect of ENSO on $O_2$ and pH in the CalCS and to address the following questions:

1. How consistent is the physical and biogeochemical response of the CalCS across ENSO events? How do these responses differ between different model resolutions?

2. What are the primary drivers and mechanisms affecting $O_2$ and pH in the CalCS?

3. Is there a difference in ENSO's influence on $O_2$ and pH between the nearshore as opposed to the offshore and between the surface as opposed to at depth?

4. How can these results help inform the observational community about the location and frequency necessary to capture ENSO signals in their time series?



This novel model setup allows us to investigate both oceanic and atmospheric components of the ENSO forcing on the CalCS, as the horizontal oceanic resolution is high enough to simulate coastally trapped waves propagating north along the coast and the atmospheric component allows for a representation of basin-scale teleconnection processes.

## 2 Model details and methods

### 2.1 Model setup and simulation

We use a prototype, fully coupled global Earth System Model (ESM2.6), developed by NOAA's Geophysical Fluid Dynamics Laboratory (GFDL). ESM2.6 was built upon the high-resolution CM2.6 physical climate model (Delworth et al., 2012; Griffies et al., 2015; Saba et al., 2016). The model's ocean component is GFDL's Modular Ocean Model Version 5 (MOM5; Griffies, 2012) with a horizontal resolution of $0.1°$. The ocean biogeochemistry model is the Carbon, Ocean Biogeochemistry and Lower Trophics (COBALT) model as used in Stock et al. (2014a, b) with modifications as described in Stock et al. (2017). COBALT includes 33 prognostic tracers and its carbonate chemistry calculation is based on the ORNL/CDIAC CO2SYS carbonate chemistry routines (Lewis and Wallace, 1998). Disk space limited the amount of output that could be saved. Therefore, we analyze all available full-depth profiles of temperature, salinity, $O_2$, and the hydrogen ion concentration ($[H^+]$), from which we compute pH, as well as surface dissolved inorganic carbon (DIC) and surface alkalinity (ALK) on monthly timescales.

We analyze a 52-year control simulation with constant 1990 atmospheric $CO_2$ forcing. As this simulation was not forced by a transient atmospheric $CO_2$ signal, it lends itself particularly well to the analysis of interannual variability. The physical climate was initialized from the beginning of year 141 of the CM2.6 1990 control simulation. The ocean biogeochemistry was initialized with modeled $O_2$, nitrate, phosphate, and silicate from the World Ocean Atlas 2005 (WOA05; Garcia et al., 2006a, b), and modeled DIC and ALK were initialized from the Global Data Analysis Project (GLODAP; Key et al., 2004). The ocean biogeochemistry in the previous ESM2.6 development version was started from a spun-up ESM2M-COBALT 1860 control simulation (Stock et al., 2014a).

### 2.2 Methods

We identified model ENSO events through the $\pm 1$ standard deviation of the wintertime (NDJ) Niño3.4 index (area-averaged SST over 5°S-5°N, 170°W-120°W). Due to drift issues in the carbonate chemistry at the beginning of the simulation, we used a Lanczos high pass filter with a cutoff frequency of 10 years (121 weights; Duchon, 1979). This procedure removed any long-term trends and decadal variability, as they were on the lower end of the frequency spectrum, and retained variability on an interannual timescale (i.e., the timescale of ENSO frequency). Using this approach, we were left with six El Niño and seven La Niña events in the time-filtered data.

For the majority of the analyses, unless otherwise noted, we employed standardized anomalies ($\sigma$, where a value of 2 corresponds roughly to a 95% significance as indicated by a Student's t-test) instead of showing absolute values. To this end, the



modeled time series was normalized at each grid point by the interannual standard deviation derived from each time series, thus allowing for a more pattern-driven interpretation of ENSO-related signals in the figures.

## 2.3 Comparison of modeled and observed ENSO in the North Pacific

To assess the model's performance in representing the physical regime of the North Pacific, we compare the physical manifes-
tation of ENSO in GFDL-ESM2.6 to a suite of observational data sets. In addition, we expand on this evaluation by comparing ESM2.6 to its precursor model, GFDL-ESM2M, which has a horizontal oceanic resolution of 1°. We use a 500-year preindus-trial ESM2M control run without transient atmospheric $CO_2$ forcing. Since we analyze normalized values, differences in the magnitude of ENSO and its impact on the CalCS ecosystem would be taken into account between the two runs, including the possible influence of different atmospheric $CO_2$ concentrations.

A comparison of climatological SST and sea level pressure (SLP) over the North Pacific reveals that both models simulate well the climatological mean SST signal and range as well as the position and magnitude of the North Pacific High pressure system off the southern US West Coast and the Aleutian Low in the Gulf of Alaska (Fig. 1a, d, and g). During El Niño (La Niña), both models represent the composite average SST signals over the whole North Pacific with positive (negative) SST anomalies along the equator and in the Gulf of Alaska and negative (positive) SST anomalies in the subtropical gyre around 30°N (Fig.
1, b, c, e, f, h, and i). Both models also simulate the intensification (weakening) of the Aleutian Low during El Niño (La Niña), though the changes are overestimated and biases in orientation relative to the observed record are apparent. It should be noted that both the orientation and position of the Aleutian Low in the observations and in ESM2.6 are likely affected by the limited number of events present in both time series and could lead to a bias in the results (Deser et al., 2017).

To shed light on the vertical structure of temperature and density along the US West Coast, we compare four vertical offshore
cross-sections from the ESM2.6 and ESM2M models to output from a Regional Ocean Modeling System (ROMS) reanalysis of the CalCS (Fig. 2). The 1980-2010 ROMS reanalysis covers the CalCS at 0.1° (∼10 km) resolution, assimilates available satellite (SST, SSH) and in situ (temperature, salinity) data (see Neveu et al. (2016) for details), and has been used extensively to describe CalCS physical dynamics, particularly in response to ENSO variability (Jacox et al., 2015b, a, 2016). ESM2.6 and the ROMS reanalysis show very similar results for the composite difference between warm and cold ENSO events, indicating
a significantly improved representation of the CalCS ENSO response in ESM2.6 relative to ESM2M. All three models agree on the sign of the temperature anomalies during El Niño (although ESM2M underestimates the magnitude of these anomalies), and ESM2.6 and ROMS highlight the largest temperature anomalies in the nearshore region and above 150 m water depth. All three models agree that density surfaces tend to deepen during El Niño (green lines) and shoal during La Niña (blue lines), with a difference in depth of up to 30 m between mean El Niño and La Niña conditions.

Additionally, we compare springtime (FMA) variability of observed versus modeled surface chlorophyll (CHL) concentrations along the US West Coast (Fig. 3). Due to its brevity, the CHL record doesn't lend itself well to an ENSO composite analysis, so we analyze CHL interannual variations via the standard deviation. We find that ESM2.6 represents well the strong cross-shore gradient in CHL variability all along the coast, with high variability of up to 2.5 standard deviations nearshore and lower variability offshore. ESM2M on the other hand only manages to reconstruct the same nearshore values in the region between

©c Author(s) 2017. CC BY 4.0 License.





40-45°N, but underestimates the cross-shore gradient significantly to the north and south of this. Both models overestimate the CHL variability offshore at around 36-40°N in comparison to the observed variability.

Finally, we compare modeled versus observed Niño3.4 indices and find that the SST variability in the tropical Pacific is over-estimated in both models, with the maximum SST variance in ESM2.6 being roughly twice as high as in ESM2M and up

to five times higher than in the observations (Fig. A1). In summary, both models simulate the large-scale atmospheric and oceanic responses associated with ENSO. However, ENSO events that are similar in magnitude to the strongest on record are far more common and regular in the model than the recent historical record suggests. This offers an opportunity to isolate large ENSO signals, though even with normalization the biogeochemical imprint of ENSO events is likely accentuated relative to other sources of variation. Furthermore, due to its high horizontal oceanic resolution, ESM2.6 does a particularly good job of

reproducing the cross-shore gradients in SST and CHL along the US West Coast, with warm temperatures and increased CHL variability close to the coast. ESM2M's coarse horizontal resolution on the other hand does not allow for a representation of finer-scale processes related to coastal upwelling and it thus fails to correctly model the cross-shore SST and CHL gradients. We thus focus on ESM2.6 to gain insight into the regional biogeochemical response of the CalCS to strong ENSO events.

## 3 Results

### 3.1 Individual representations of ENSO in ESM2.6 in the CalCS

The SST and SLP responses of the CalCS to six different modeled El Niño events are highly variable (Fig. 4). The SST and SLP signals in Fig. 4a are the most similar to the signals typical during an Eastern Pacific-type El Niño in the observational record (e.g., Alexander et al., 2002; Capotondi et al., 2015). In this case, the CalCS experiences positive SST anomalies within ∼200-300 km of the coast and negative SST anomalies further offshore in the region of the subtropical gyre, both with a

magnitude of $\pm 2\sigma$. This reflects the typical SST signal in the eastern North Pacific prevalent during El Niño with a strong cross-shore SST gradient. In addition, the SLP signal reflects the typical atmospheric pattern over this region associated with El Niño, with negative SLP anomalies of up to -2$\sigma$ to the north of ∼38°N, indicating an intensification of the Aleutian Low over the Gulf of Alaska, and positive SLP anomalies of around 1$\sigma$ to the south. Although the SST anomalies in Fig. 4c are similar to Fig. 4a, the SLP anomalies are substantially different with negative SLP anomalies over the whole CalCS region and

the adjacent landmass. The other cases show more widespread negative SST anomalies closer to the coast of 1-2$\sigma$ (Fig. 4b and d), a more intense warming in the Gulf of Alaska (Fig. 4e), and stronger, more domain-wide positive SST anomalies with a maximum greater than 2$\sigma$ offshore of the Oregon coast (Fig. 4f). During La Niña, the SST and SLP anomalies tend to be of opposite sign, with cold anomalies close to the coast and warmer waters further offshore (Fig. A2).

Figure 5 shows the temporal evolution of surface and subsurface (100 m) temperature, $O_2$, and pH for the six individual El

Niño and seven individual La Niña events, as well as the composite mean, for a region within 100 km of the coast and averaged over 34–44°N (see Fig. 5g for the specified region). We analyze the surface as well as 100 m since the surface is where heat transfer and chemical interactions with the atmosphere take place, and thus the atmospheric impact of ENSO is the greatest, and 100 m is roughly the depth of the heart of the thermocline.





At the surface, temperature anomalies are on average positive (negative) during El Niño (La Niña) with values of $-1\sigma$ to $3\sigma$ ($-2\sigma$ to $1\sigma$) for the individual events (Fig. 5a and d). In the case of $O_2$, the coastal surface waters experience lower-than-usual $O_2$ conditions during El Niño ($\sigma$ on average around -1), whereas $O_2$ concentrations tend to be elevated with $\sigma$ of up to 1 during La Niña (Fig. 5b and e). At 100 m, the strongest temperature signal occurs during the spring (FMA) following the typical

El Niño season (NDJ). Furthermore, the CalCS experiences higher-than-usual $O_2$ conditions during El Niño and lower $O_2$ concentrations during La Niña at 100 m. As was the case with temperature, the largest $O_2$ signal occurs with $\sigma$ between 1 and 2 during FMA. This contrasting behavior exhibited by $O_2$ at the surface and at 100 m is not exhibited by pH, which like temperature displays a consistent signal throughout the water column (Fig. 5c and f).

For all three variables, the magnitude of the standardized anomalies is very comparable both during El Niño and La Niña. Overall, the variability between the individual warm and cold events in Figs. 4 and 5 is quite large, indicating on the one hand the variability in the atmospheric and oceanic responses to tropical Pacific conditions and on the other hand the importance of internal climate variability in driving anomalies unrelated to ENSO (e.g., Deser et al., 2017). Furthermore, the initial conditions differ substantially from event to event and thus the temperature, $O_2$, and pH responses could be in part controlled by the mean

background state of the ocean immediately preceding an event.

### 3.2 Response of temperature, $O_2$, and pH to mean ENSO signal

We illustrate the mean response of surface temperature, $O_2$, and pH to ENSO in Fig. 6a-c, where we show standardized anomalies of these quantities, representing El Niño minus La Niña composites. Showing warm minus cold anomalies amplifies the ENSO signal, thus increasing its significance. This approach assumes linearity in the signal, i.e., that the influence of El

Niño is -1 times the influence of La Niña. SST anomalies here are positive between the coast and $\sim$300-500 km offshore, as well as offshore north of 40°N and south of 28°N, with anomalies of up to $2\sigma$ (Fig. 6a). Offshore between $\sim$28°N and 40°N, in the region of the subtropical gyre, SST anomalies are negative around $1$-$2\sigma$, and reflect a typical observed El Niño signal. The surface $O_2$ anomalies largely reflect the SST signal, albeit with a reversed sign, as would be expected from a solubility-driven response. Close to the US West Coast, $O_2$ anomalies are lower than $-2\sigma$ and become less negative further offshore. Positive

$O_2$ anomalies of around $0.8$-$2\sigma$ are found offshore in the region of the subtropical gyre (Fig. 6b). The surface pH signal differs from the SST and $O_2$ signals, in that positive pH anomalies, with a maximum around $2\sigma$, are limited to a very narrow band of $\sim$100 km along the coast (Fig. 6c). Between 100 km and around 500 km offshore, pH anomalies are negative around $-0.8\sigma$, while further offshore in the region of the subtropical gyre, pH anomalies are largely positive again.

We investigate ENSO-driven changes in temperature, $O_2$, and pH in the subsurface (100 m) in Fig. 6d-f. A similar cross-shore

gradient in temperature occurs at 100 m compared to the surface, with anomalies larger than $2\sigma$ in some regions along the coast (Fig. 6d). However, the 100 m signal is more limited to within a $\sim$100 km wide band along the coast. This difference can be explained by coastally trapped waves depressing the thermocline on their path poleward along the coast, by weakened upwelling along the CalCS coast, likewise affecting the depth of the thermocline, or by a combination of both processes. At the surface on the other hand, warming through the atmosphere likely contributes to a broader area of positive anomalies.





The mean response of 100 m $O_2$ to ENSO is drastically different than the $O_2$ response at the surface, with positive anomalies within a 100 km band along the coast, indicating that two different processes govern the $O_2$ response to ENSO at the surface and at depth (Fig. 6e). The 100 m pH signal looks largely the same as the surface pH signal, with a slightly broader cross-shore region with positive pH anomalies of up to $2\sigma$ (Fig. 6f). These positive pH anomalies are also more widespread northward and

southward along the coast at 100 m compared to the surface. Furthermore, the 100 m pH signal is very similar to the 100 m $O_2$ signal (compare Fig. 6e and 6f), suggesting a common subsurface process influencing both variables.

In Fig. 7 we examine the same offshore cross-sections as in Fig. 2 and focus on the response of temperature, $O_2$, and pH to ENSO in the vertical plane. The differing response of $O_2$ to ENSO at the surface and at 100 m that we observed in Fig. 6 is also clearly visible in all four offshore cross-sections in Fig. 7 (b, e, h, and k). The $O_2$ and pH responses in Fig. 7(c, f, i, l) are

very similar between 30°N and 35°N, suggesting that here, they are both affected by the same processes. Between 40°N and 45°N on the other hand, pH anomalies are largely positive throughout the water column, indicating that in this region there is a different process at play affecting pH in the top ∼100 m as opposed to $O_2$ at the same latitude.

To disentangle the processes and drivers behind these changes in $O_2$ and pH and ultimately understand how ENSO affects these quantities in the CalCS, we next split the $O_2$ and pH El Niño composite means into their individual components.

## 3.3    Drivers and processes behind changes in $O_2$ and pH in the CalCS

In the surface ocean, $O_2$ is affected by four main processes: (i) changes in SST, which affect the $O_2$ solubility, (ii) wind-driven variability, which drives the $O_2$ gas exchange across the air-sea interface, (iii) primary production, which reduces $CO_2$ and increases $O_2$ in the surface ocean, and (iv) ocean circulation, which affects horizontal advection and upwelling. We assume that the total change in $O_2$ during El Niño is the sum of two components that include temperature-related and temperature-

unrelated processes (after Fay and McKinley, 2013), thus:

$$\Delta O_2 \approx \underbrace{\frac{\partial O_2}{\partial T} \cdot \Delta T}_{\text{temperature-related}} + \underbrace{\text{Residual}}_{\text{temperature-unrelated}} \tag{1}$$

For the analysis in Fig. 8, we derive the temperature-related component from the solubility of $O_2$, using the solubility coefficients of Weiss (1970) as demonstrated in Sarmiento and Gruber (2006). Since the total change in $O_2$ is just the composite mean ($\Delta O_2$ in Equation 1), we calculate the temperature-unrelated component, or residual term, from the difference between

$\Delta O_2$ and the temperature-related component. The negative correlation between temperature and $O_2$ in the surface ocean (as seen in Fig. 6a and b) can largely be explained by a rise in temperature causing a decrease in the amount of $O_2$ that can be dissolved in the surface waters and thus leading to a drop in surface $O_2$ (Fig. 8b). We argue that while increased storm activity during El Niño could enhance the $O_2$ gas exchange through the air-sea interface, the decrease in wind stress over the CalCS, which is typically associated with El Niño, is expected to have an overall reducing effect on the $O_2$ gas exchange. As the CalCS

is on average a source of $O_2$ to the atmosphere (Sarmiento and Gruber, 2006), this effect contributes to an increase in $O_2$, and thus opposes the solubility effect. The residual effect (Fig. 8c), which is a combination of upwelling of waters with a certain $O_2$ signal and the biological imprint on $O_2$ concentrations, acts on average to decrease $O_2$ in the surface ocean. This is likely





due to primary production being limited during El Niño (e.g., Bograd and Lynn, 2001).

At 100 m, where El Niño leads to an increase in $O_2$ along the coast and up to $\sim$200 km offshore, the main driver of the $O_2$ change is the residual term, i.e., the contribution that is not directly linked to changes in temperature (Fig. 8f). This term could include local biological effects and/or circulation-driven changes between water masses with different histories of $O_2$ supply

and consumption. In this case, the simulated $O_2$ increase is consistent with the deepening of isopycnal surfaces during El Niño (Fig. 2), replacing older, $O_2$-poor waters associated with dense subsurface waters in the CalCS, with less dense, elevated $O_2$ waters. Figure 9 demonstrates that during El Niño, the depth at which oxygen falls below the hypoxic threshold (where $O_2$ is $\leq$60 $\mu$mol kg$^{-1}$) deepens on average by 20 m in the first 100 km along the coast from a mean climatological depth of around 300 m, representing a 6-7% change. This deepening of $O_2$-depleted waters, where remineralization rates are high, can help

explain the modeled increase in $O_2$ seen along the coast in Fig. 7 (b, e, h, and k) and Fig. 8d.

We consider the contributions of surface DIC, ALK, temperature (T) and salinity (S) to ENSO-driven changes in surface pH in Fig. 10. For this analysis we focus on surface values, as DIC and ALK were saved only in the surface layer in the simulation. To decompose pH into these four drivers, we use a Taylor expansion according to the following equation (after Lovenduski et al., 2007; Doney et al., 2009b; Turi et al., 2014, 2016):

$$\Delta \text{pH} \approx \frac{\partial \text{pH}}{\partial \text{DIC}} \cdot \Delta \text{DIC} + \frac{\partial \text{pH}}{\partial \text{ALK}} \cdot \Delta \text{ALK} + \frac{\partial \text{pH}}{\partial \text{T}} \cdot \Delta \text{T} + \frac{\partial \text{pH}}{\partial \text{S}} \cdot \Delta \text{S} \qquad (2)$$

where the partial derivatives denote the sensitivities of pH to small changes in the four drivers. We determined these sensitivities using the online tool "CO2calc" (Robbins et al., 2010) and then multiplied each of them by the change in each variable as calculated in the ENSO composites shown in Fig. 8 (corresponding to the $\Delta$-terms in Equation 2).

Changes in DIC are primarily what drive the increase in pH in the nearshore 100 km of the central CalCS (34°N–40°N; Fig.

10c), whereas the contributions from SST and ALK counteract the effect of DIC by reducing pH in the same region (Fig. 10b and d). During El Niño, upwelling is weakened and the thermocline is depressed, thus limiting the supply of low-pH and high-DIC waters to the surface. Likewise, the supply of cold, high-ALK waters is inhibited, leading to a decrease in pH due to a positive correlation with ALK and a negative correlation with temperature. North of $\sim$40°N, changes in DIC and SST contribute to an overall slight decrease in pH, whereas changes in ALK lead to an increase in pH. This result suggests that

there is a dipole pattern in the upwelling along the coast, with a decrease in upwelling in the central CalCS and a potential increase in the northern CalCS during El Niño. Thus, the SST contribution to pH seems to mainly act through the mechanism of surface heat fluxes, rather than through changes to the upwelling of cooler waters, as the contribution of changes in SST to the overall changes in pH are of the same sign all along the coast and extend further offshore than the contributions of both DIC and ALK. The contributions of salinity and of the residual term are negligible throughout the whole CalCS (Figs. 10e and

f).



## 4    Discussion and Conclusions

The influence of ENSO on the physics and ultimately the biogeochemistry of the CalCS is very complex, due to the multiple processes that are affected by ENSO-driven climate variability. In this study, we delved into the processes through which ENSO can influence $O_2$ and pH in the CalCS, and explained how physical variability can influence the carbon and oxygen systems

of the CalCS. Our results demonstrate that in the surface ocean above the thermocline (above $\sim$100 m), interannual variability associated with ENSO modulates $O_2$ and pH through two different mechanisms. In the case of surface $O_2$, the strongest signal extends several 100 km offshore, mirroring the SST signal where elevated temperatures during El Niño lower the $O_2$ solubility and thus cause a decrease in $O_2$. The decomposition of $O_2$ into temperature-related and temperature-unrelated components further confirmed that changes in the $O_2$ solubility are the main driver for the surface $O_2$ anomalies. Furthermore, our results

show a decoupling between the response of $O_2$ to ENSO in the surface ocean above the thermocline and the waters below that. Below 100 m, $O_2$ is mainly modulated through changes in the vertical structure which affect the depth and location of the hypoxic threshold. During El Niño for example, while the surface ocean above the thermocline experiences a decrease in $O_2$, the concentration of $O_2$ in the waters below that increases and the hypoxic threshold deepens due to a depression of the thermocline. During La Niña, the mechanism is reversed and we model an increase in $O_2$ at the surface and a decrease

below 100 m (not shown). In the case of pH on the other hand, the strongest changes occur in and are limited to a narrow band of $\sim$100 km along the central US West Coast ($\sim$34–40°N), both at the surface and below the thermocline. We inferred from the decomposition of pH into its individual components that changes in DIC, driven by modifications to the depth of the thermocline during an ENSO event, are the main driver of pH variability in the nearshore 100 km, whereas temperature and ALK counteract the DIC effect. Further offshore on the other hand, SST changes become more important in determining pH

variability and the contributions of DIC and ALK tend to cancel each other.

These results also suggest that during La Niña, the first 50-100 km along the coast are more readily supplied with cool, nutrient-rich water, thus enhancing the coastal upwelling effect, and potentially fueling biological production. At the same time, this process also brings more $[H^+]$, DIC, and ALK to the surface, thus overall lowering surface pH more toward a state of acidity. During El Niño on the other hand, this process is suppressed, thus limiting the supply of nutrients and inhibiting biological

production. At the same time however, the supply of $[H^+]$ is also limited, therefore having an overall increasing effect on surface pH. These findings for pH are in line with Nam et al. (2011), who noted an increase in nearshore surface pH during El Niño and a decrease during La Niña due to an uplifting of isopycnal surfaces and thus a higher supply of low-pH waters to the surface. In the case of $O_2$ on the other hand, our results differ from what Nam et al. (2011) concluded. While we found a significant increase (decrease) in $O_2$ in the nearshore surface waters above 100 m during La Niña (El Niño), their study

suggests that during the 2010/2011 La Niña, $O_2$ anomalies were lower than average. Their explanation for this observation is that due to an increased poleward flow of the California Undercurrent during La Niña, subsurface primary production was limited and thus caused a reduction in $O_2$ concentrations. Our results however suggest that the modeled increase (decrease) in surface $O_2$ during La Niña (El Niño) is mainly driven by changes to the $O_2$ solubility due to anomalous surface cooling (warming). Furthermore, the analysis suggests that below 100 m, the opposing decrease (increase) in nearshore $O_2$ during




La Niña (El Niño) is attributable to a shoaling (deepening) of isopycnal surfaces, affecting the location of low-$O_2$ waters. In addition to this analysis, the role of horizontal advection in the model needs to be investigated in more detail, but is beyond the scope of this study.

As Nam et al. (2011) pointed out, the large variability in $O_2$ and pH during ENSO events seen in their study, which can be seen also in ours, suggests that the carbonate ion concentration ($[CO_3^{2-}]$) and thus the saturation state of aragonite ($\Omega_{arag}$) experience similar fluctuations in their amplitude (Turi et al., 2016). If the pH drops below 7.75, which corresponds to $\Omega_{arag} < 1$, waters become unfavorable for calcifying organisms to build and maintain their shell structure (e.g. Feely et al., 2008; Bednaršek et al., 2014; Bednaršek and Ohman, 2015). Our results suggest that during La Niña, the surface waters are more corrosive but but also more oxygenated than during El Niño, while the waters below 100 m are both more corrosive and more deoxygenated. While primary production in the CalCS is greater during La Niña, the ecosystem is also more stressed by $O_2$-, low-pH, and low-$\Omega_{arag}$ conditions which could add to or enhance the impact of these stressors due to climate change, as also noted by Nam et al. (2011).

This modeling study can help inform future observational studies on the frequency as well as horizontal and vertical resolution of $O_2$ and pH measurements necessary to capture the surface and subsurface biogeochemical expressions of ENSO in the CalCS. We show that it is critical to have a strong observational network particularly in the nearshore 100 km along the US West Coast, as this is where the ENSO signal is largest, both in $O_2$ and in pH. Furthermore, the modeled contrasting response of $O_2$ to ENSO at the surface and at 100 m highlights the necessity of having vertical sections that go deep enough and have a sufficient vertical resolution to capture signals both in the euphotic zone above the thermocline as well as in the waters below it. This study additionally demonstrates that the diversity of ENSO events might contribute to the variability of the physics and biogeochemistry in the CalCS, and thus emphasizes the importance of analyzing long enough time series to include a variety of different events. In addition to the variability between different types of ENSO events, other sources of internal climate variability unrelated to ENSO contribute to noisiness in the physical and biogeochemical signals in the CalCS (e.g., Deser et al., 2017).

*Data availability.* The ROMS reanalysis output is available from http://oceanmodeling.ucsc.edu. The GFDL model output is available upon request. The SLP observational data are from the NCEP Reanalysis and were downloaded at https://www.esrl.noaa.gov/psd/data/gridded/. The SST observational data are from the Hadley Center at http://www.metoffice.gov.uk/hadobs/hadisst/. Finally, the CHL data are from SeaWIFS 1998-2010 and were obtained from NASA at
https://oceandata.sci.gsfc.nasa.gov/SeaWiFS/.

*Author contributions.* GT wrote the manuscript. GT, MA, CS, and JD outlined the initial stages of the project. GT, MA, NSL, AC, and JS were responsible for streamlining the direction and scope of the project. JS created all figures except for Fig. A1 (created by GT). MJ supplied the ROMS simulation output. CS, JD, and JJ set up and ran the ESM2.6 and ESM2M models. JJ managed the model output and storage. All authors contributed to the discussion and revision of the manuscript contents.



*Competing interests.* The authors declare that they have no conflict of interest.

*Acknowledgements.* The authors thank Dr. Matthew Newman from NOAA/ESRL for his support and help in outlining the initial stages of the project. All authors are grateful to the support from NOAA's Climate Program Office through the Marine Tipping Points program. NSL is grateful for support from NSF (OCE-1558225).



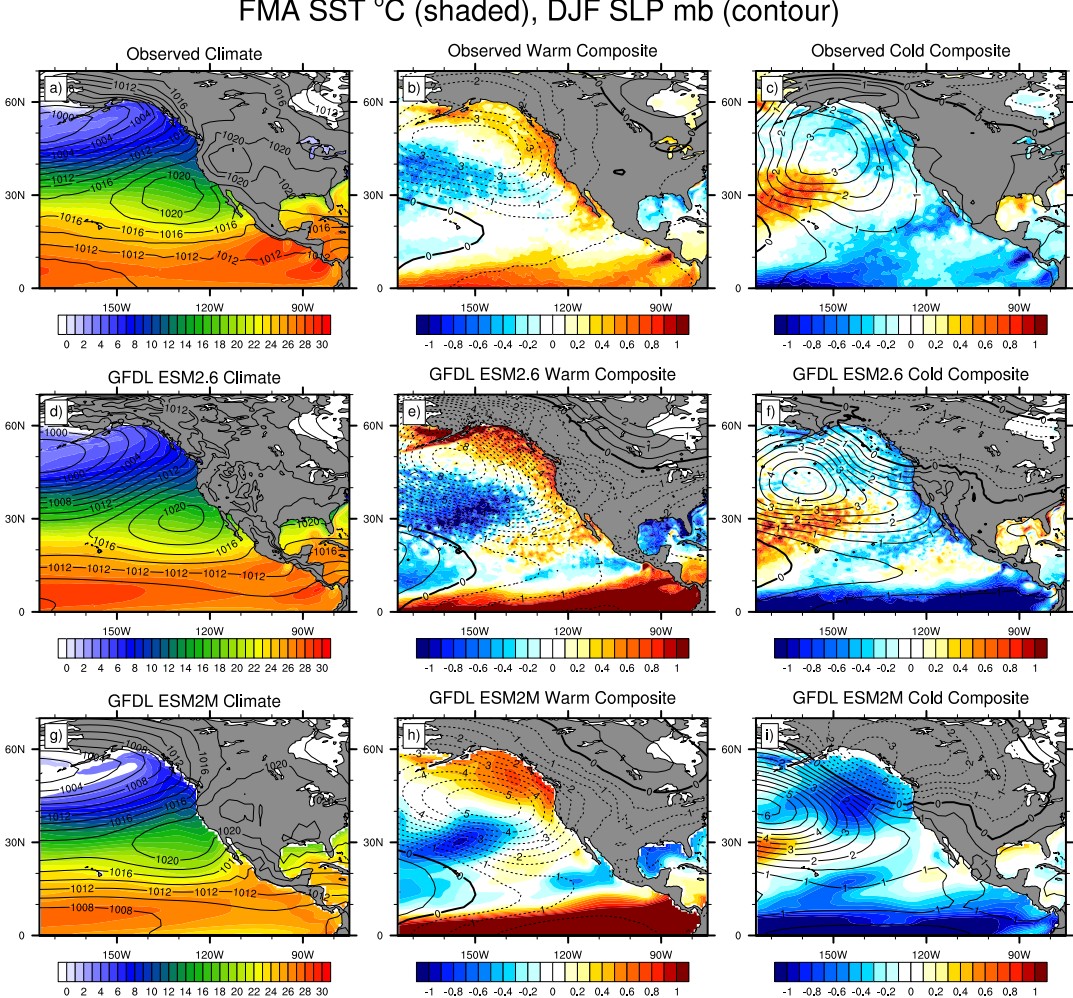

**Figure 1.** FMA SST (°C, shaded) and DJF SLP (mb, contours) in the CalCS: (a, d, and g) climatology (4 mb contour interval), (b, e, and h) El Niño/warm high-pass composite (0.5 mb contour interval), and (c, f, i) La Niña/cold high-pass composite (0.5 mb contour interval) for (a-c) observations (HadISST, NCEP Reanalysis), (d-f) GFDL-ESM2.6, and (g-i) GFDL-ESM2M.






**Figure 2.** GFDL-ESM2.6 depth-longitude cross-sections of FMA warm-cold high-pass filtered standardized anomalies of temperature for (first column) ESM2.6, (second column) ESM2M, and (third column) ROMS climatology. The contours show the mean climatological density lines (first row) 44°N, (second row) 40°N, (third row) 36°N, and (fourth row) 32°N for the 222 km nearest the US West Coast. The bold lines show the positions for the σ=25 level (ESM2.6 and ESM2M) and the σ=25.5 level (ROMS) for the climatology (black), El Niño conditions (green) and La Niña conditions (blue) and the thin lines indicate density increments of 0.25.





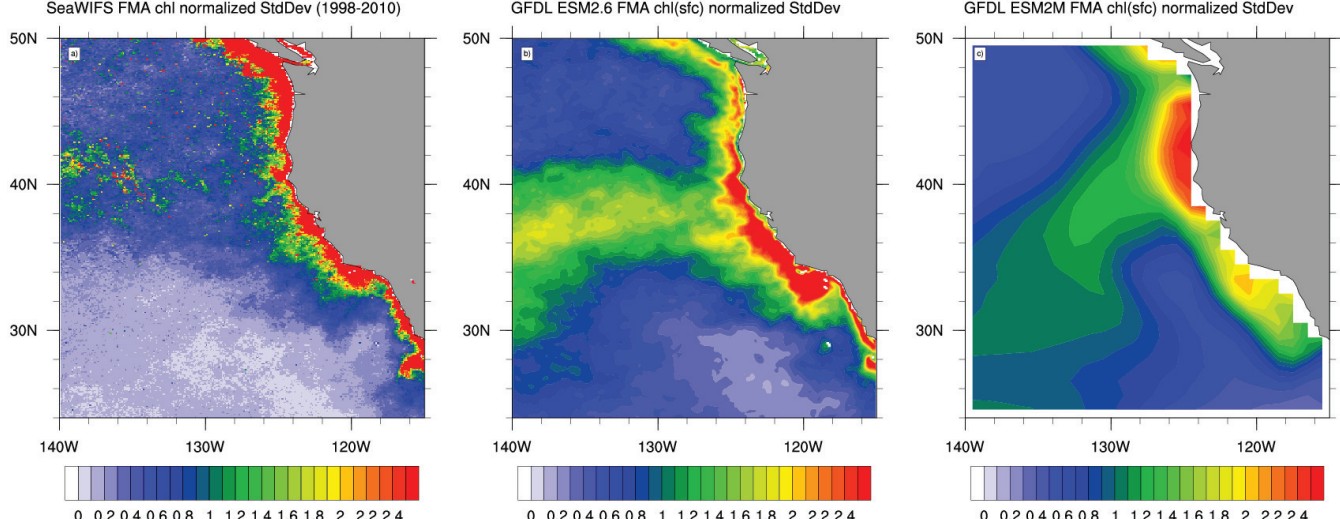

**Figure 3.** Springtime (FMA) interannual standard deviation of surface chlorophyll concentrations (normalized by area average to produce comparable scales) for a) 1998-2010 NASA-SeaWIFS observations, b) GFDL-ESM2.6, and c) GFDL-ESM2M.





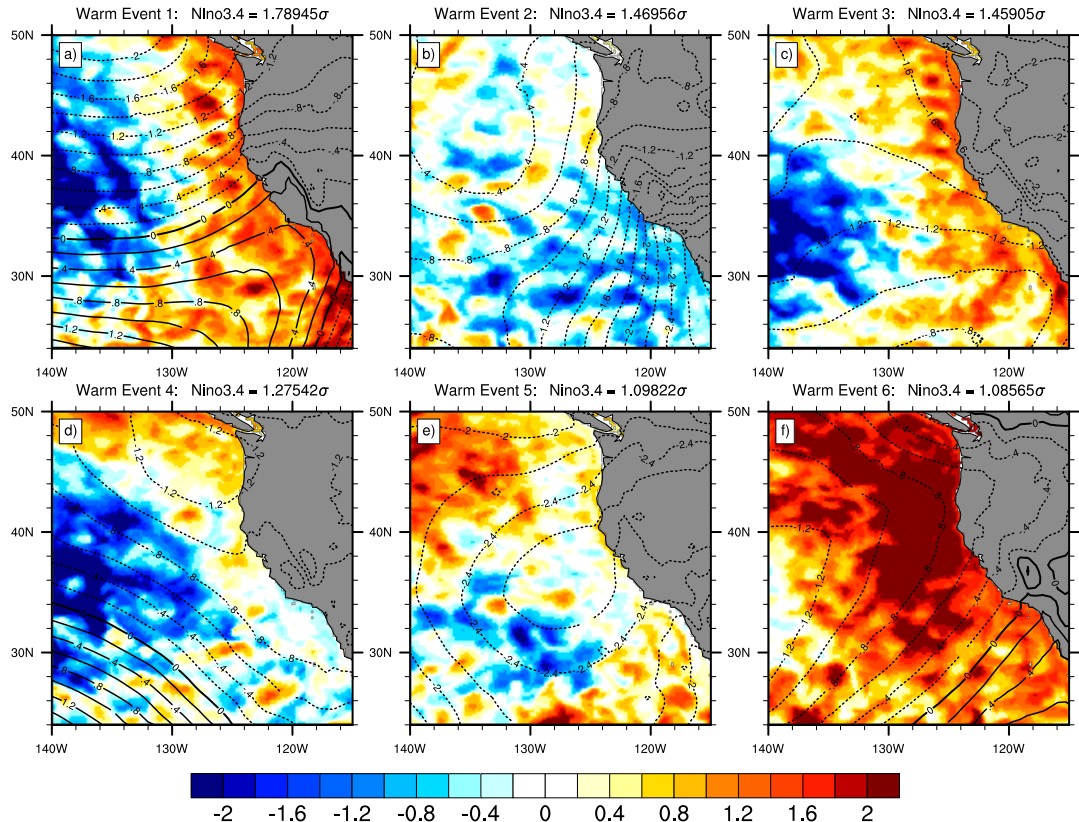

**Figure 4.** GFDL ESM2.6 FMA high-pass filtered standardized anomalies for SST (shaded) and SLP (0.2 $\sigma$ interval contours) for the top six El Niño events based on NDJ Niño3.4 anomalies.



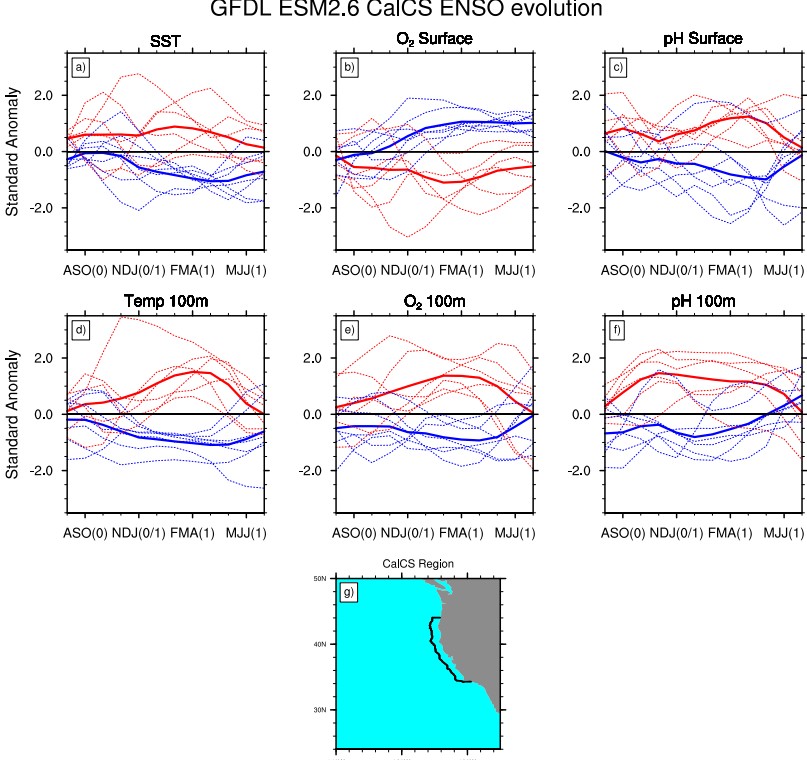

**Figure 5.** GFDL ESM2.6 temporal evolution of individual warm (red) and cold (blue) events (dashed) and composite of all events (solid) for area averages between $34°N$ and $44°N$ and $100\,km$ offshore of the US West Coast (see gray boxes in Fig. 6) for a) SST, b) surface $O_2$, c) surface pH, d) $100\,m$ temperature, e) $100\,m$ $O_2$, and f) $100\,m$ pH. The time series are high pass filtered and seasonally averaged and show the evolution of standardized anomalies during the peak ENSO years from JAS before the peak to JJA after the peak. The gray outline in g) highlights the area within the CalCS used for this analysis.



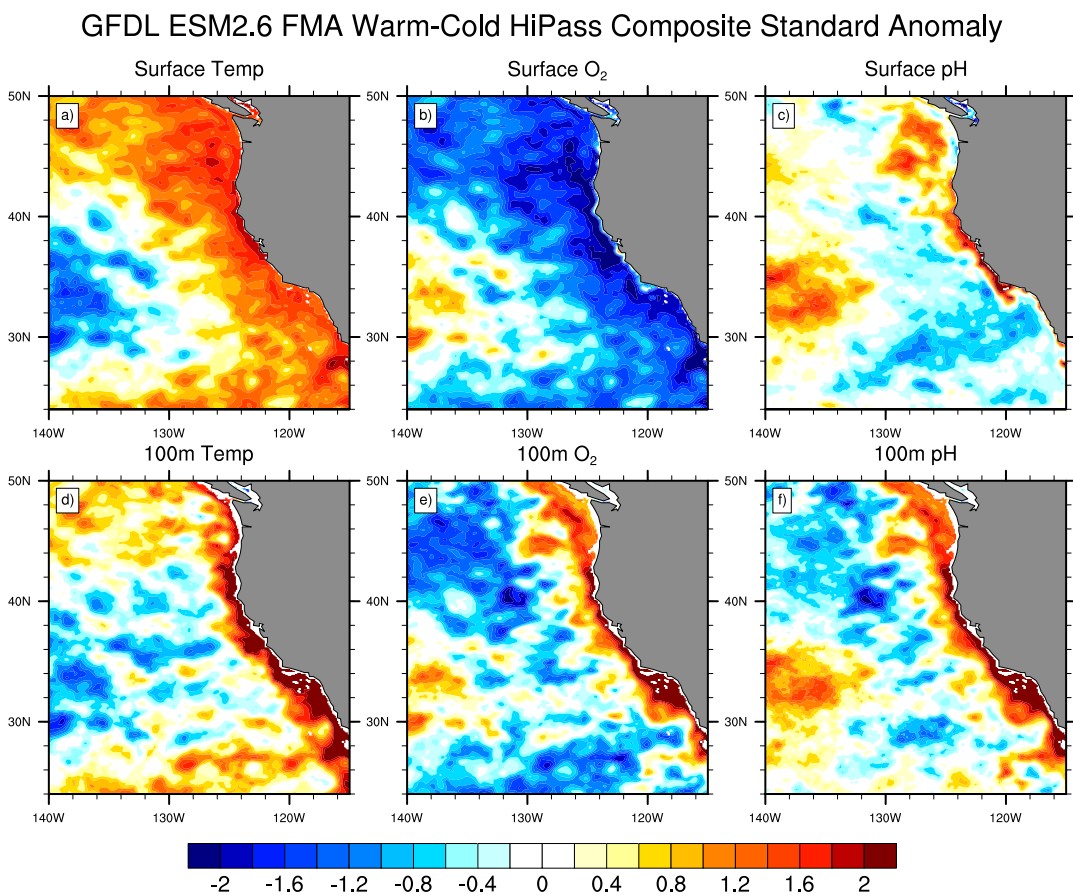

**Figure 6.** GFDL-ESM2.6 FMA warm-cold high-pass filtered standardized anomalies for a) SST, b) surface $O_2$, c) surface pH, d) 100 m temperature, e) 100 m $O_2$, and f) 100 m pH.

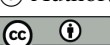


**Figure 7.** GFDL-ESM2.6 depth-longitude cross-sections of FMA warm-cold high-pass filtered standardized anomalies of (first column) temperature, (second column) O$_2$, and (third column) pH (shaded) with mean climatological density lines (contours) along (first row) 44°N, (second row) 40°N, (third row) 36°N, and (fourth row) 32°N for the 222 km nearest the US West Coast. The bold lines show the positions for the $\sigma$=25 level for the climatology (black), El Niño conditions (green) and La Niña conditions (blue) and the thin lines indicate density increments of 0.25.





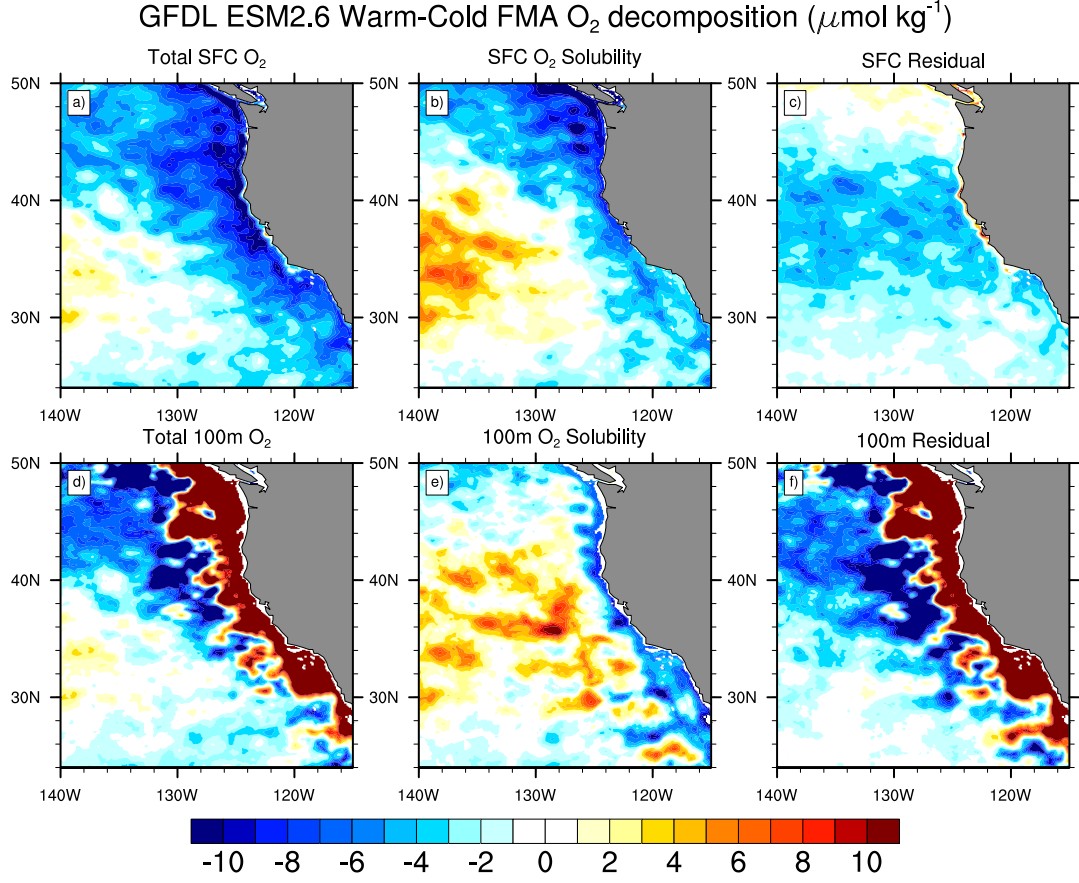

**Figure 8.** GFDL-ESM2.6 FMA warm-cold high-pass filtered anomalies for $O_2$ and its components a) surface total $O_2$, b) $O_2$ due to $\Delta$ surface temperature (solubility), c) residual surface processes, and d) 100 m total $O_2$, b) $O_2$ due to $\Delta$ 100 m temperature (solubility), c) residual 100 m processes.




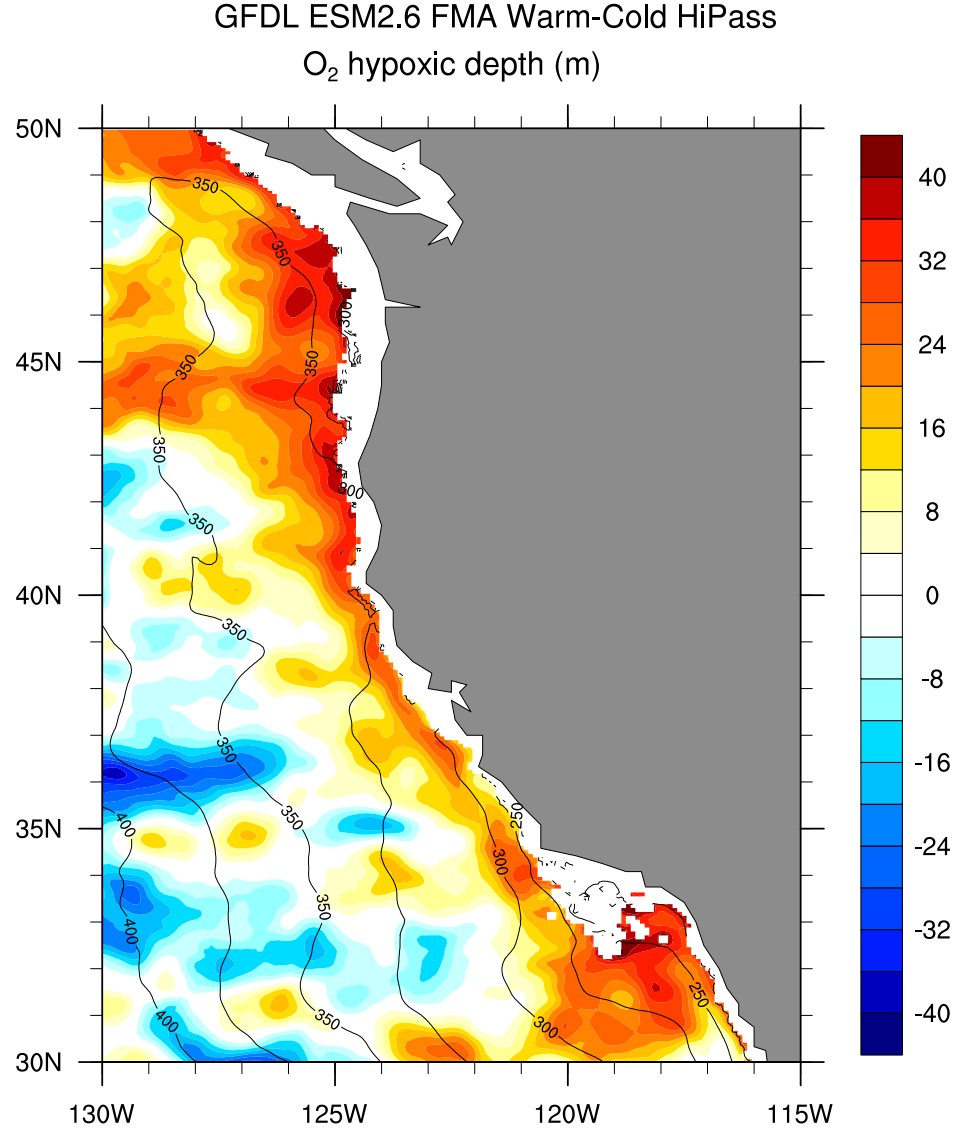

**Figure 9.** GFDL ESM2.6 FMA warm-cold high-pass filtered anomalies (m) for depth of the hypoxic threshold ($O_2 \leq 60\ \mu\mathrm{mol\,kg^{-1}}$) and FMA mean climatological depth of the hypoxic threshold (25 m interval contours). White areas near the coast indicate an absence of the hypoxic level in the vertical layer. Positive numbers indicate a deepening of the hypoxic threshold.



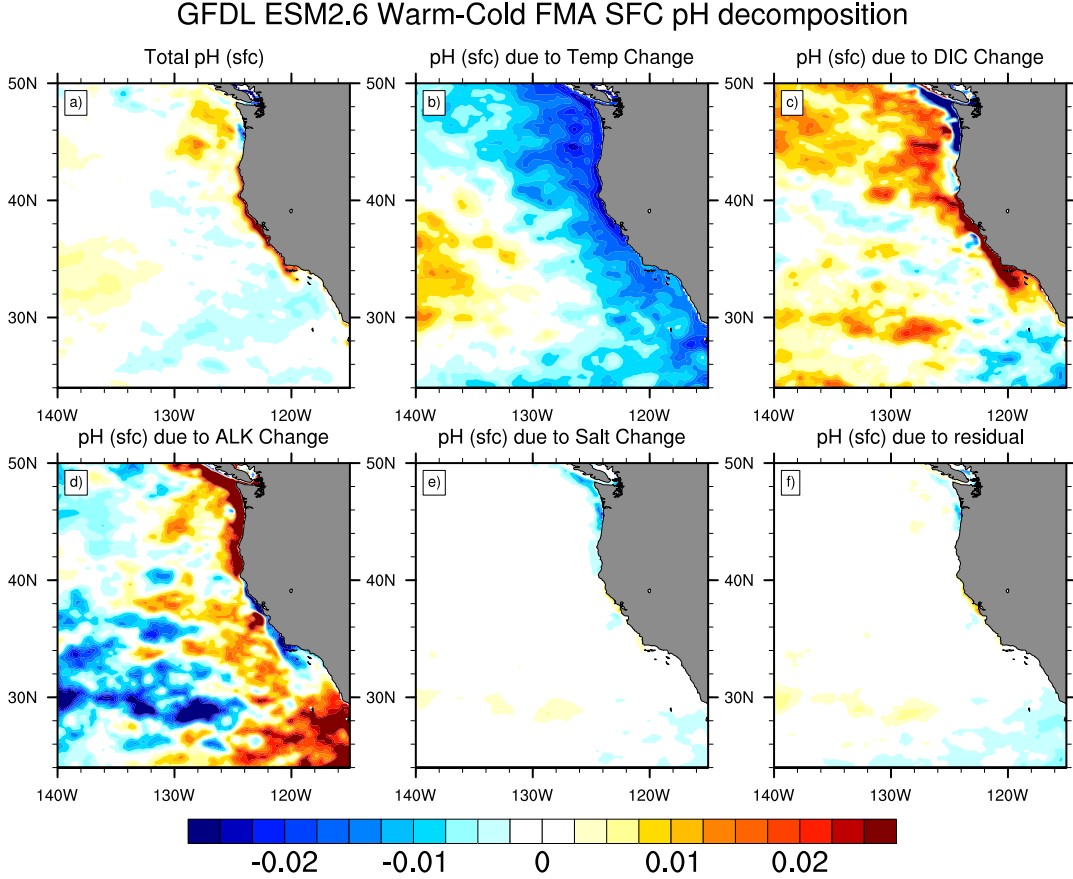

**Figure 10.** GFDL-ESM2.6 FMA warm-cold high-pass filtered standardized anomalies for surface pH and its components a) total pH, b) pH due to $\Delta$ temperature, c) pH due to $\Delta$ dissolved inorganic carbon, d) pH due to $\Delta$ total alkalinity, e) ph due to $\Delta$ salinity, and f) residual processes.







**Figure A1.** Left: Spectral analysis of Niño3.4 indices for ESM2.6 (blue), ESM2M (green) and observations (black; CPC NOAA). The shaded green area represents the $\pm 1$ standard deviation range of ten 50-year segments of the 500-year ESM2M control run. Right: corresponding color-coded Niño3.4 index time series. Note that the three x-axes all show 52 years to be comparable with the whole ESM2.6 time series (ESM2.6: model years 1-52; ESM2M: model years 449-500; observations: years 1963-2014).



## GFDL ESM2.6 FMA SST(shaded) SLP(contour) Hipass Seas StdAnom

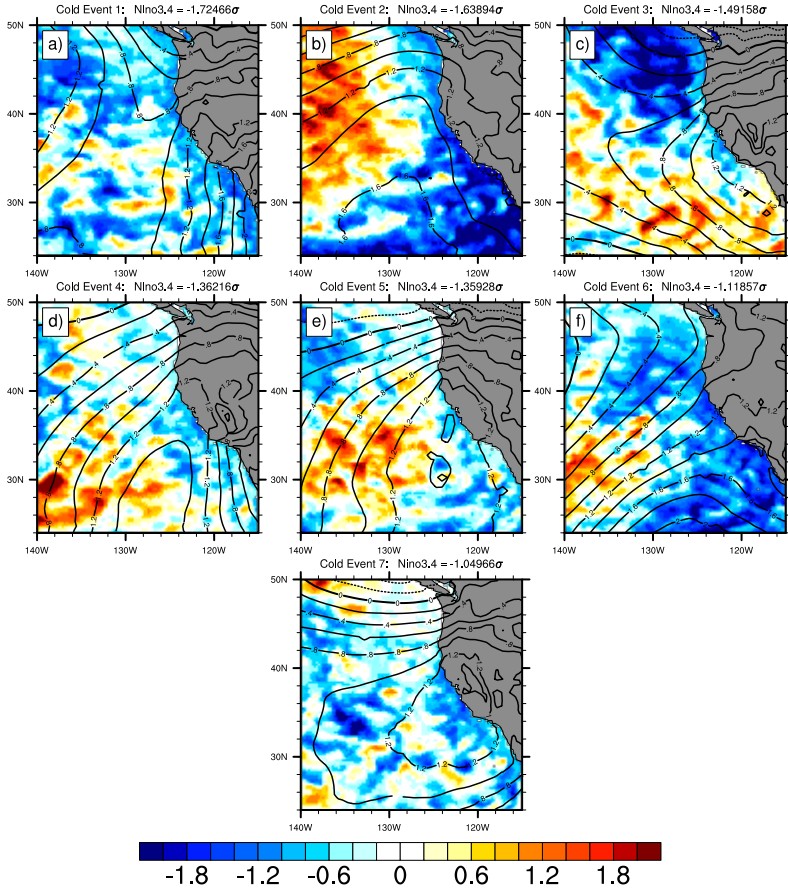

**Figure A2.** GFDL ESM2.6 FMA high-pass filtered standardized anomalies for SST (shaded) and SLP (0.2 $\sigma$ interval contours) for the top seven La Niña events based on NDJ Niño3.4 anomalies.



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
