# Peer review of "Response of O₂ and pH to ENSO in the California Current System in a high resolution global climate model"

_Ocean Science, 2017_

## Referee Comment (RC1) · Anonymous Referee #1 · 30 Aug 2017

**1. Summary**

Turi et al. use a high-resolution fully-coupled global earth system model (GFDL-ESM2.6) to uncover the O2 and pH response of the California Current System (CalCS) to El Niño/Southern Oscillation (ENSO). Despite significant variations of the response to individual events due to ENSO diversity, composite means for warm/cold events reveal consistent physical and biogeochemical changes along the US West Coast. While the O2 response to ENSO is wide-spread and differs between the surface (driven by changes in the solubility) and at 100m (driven by changes in the thermocline structure), the pH response is mainly confined to the coastal environment, highlighting the dominant role of changes in dissolved inorganic carbon (DIC) and upwelling associated with ENSO.

**2. General comments**

The influence of ENSO on the physical and biogeochemical environment of the CaICS is evidenced both in observational and model-based data. Investigating the associated changes in the current system improves our understanding of the ecosystem functioning, its sensitivity to change and is relevant for ocean management. Even though the imprint of ENSO on the CaICS has been thoroughly studied, the diversity among ENSO events and the lack of high-frequency 3-dimensional observational records render modeling studies (such as the one presented) extremely valuable in providing opportunities to study ecosystem changes on interannual time scales. The paper presents results on O2 and pH changes in the CaICS based on a global high-resolution fully coupled earth system model, which is, to my knowledge, unprecedented. In my opinion, it thus addresses relevant scientific questions within the scope of Ocean Science. The work is generally well presented and structured. I recommend this manuscript for publication with Ocean Science after addressing the few minor comments on the manuscript listed below.

3. Specific and purely technical comments

**Methods**

Page 5, Line 1: What do you mean by "interannual standard deviations"? I assume you apply the Lanczos filter as described, deseasonalize the data which yields time series that retain only anomalies on interannual time scales, right? From these time series, you compute standard deviations and use them to normalize the response? I guess the term "interannual standard deviations" got me confused and you might want to clarify this passage, as the normalization of all the data is key to interpreting your results

Model Evaluation

The authors demonstrate that ESM2.6 shows an improved coastal ENSO response compared to ESM2M. They also compare their model results to a data-assimilative ROMS hindcast. Can the authors elaborate on the reason of using the data-assimilative regional model for means of comparison? Would e.g. the long (though not as high-frequent) CalCOFI records provide an additional opportunity for a model-independent evaluation of the presented response?

Page 6, Lines 5-10: Nino 3.4 variance seems substantially overestimated in the global models. Moreover, ENSO seems to be more periodic with a maximum in the frequency spectrum at 3 years return time (Figure A1). I would like to encourage the authors to elaborate on this fact and the potential implications of this particular issue in the model for their interpretation of the presented results in more detail.

Figure 5: Plot a) showing SSTs could basically be backed up by satellite observations, not? I was surprised to see such a large spread among different events, in particular in the phasing of the peak anomalies (is this ESM specific, or does it really reflect the ENSO diversity?). It would be interesting to see whether observations show similar differences, or whether the phasing is more synchronous.

Page 7, Line 19: Just out of curiosity, why is it that the ESMs do not reproduce the asymmetry in Nino3.4 indices we find in observational records?

**Results**

Page 7, Line 32 and Page 8, Line 6: I think it is important to be more specific about the "subsurface process" that is likely dominating the response you observe at 100m. Using a hindcast simulation on a regional model setup covering the period from 1979-2016, Frischknecht et al. 2017 discussed forcing mechanisms that are also relevant for the findings you present (see their Figure 4). They showed that the bulk part of the coastal response to ENSO is due to changes in the density structure of the water column. While these changes are mainly driven by oceanic forcing (i.e. through coastally trapped waves), changes in the wind forcing do not explain the deepening of isopycnals during El Nino, but cause changes in upwelling velocities.

Page 8, Line 11: Maybe worth remembering the reader that you are discussing EN-LN differences. I got confused here when I first read through it and wasn't sure what this "largely positive" refers to.

Page 9, Line 25: This "potential upwelling increase in the northern CalCS during El Nino" seems puzzling to me. What forcing mechanisms would actually be the cause? I am not aware of other studies that would support this finding, and if there are, please refer to them. I think backing up this finding, if possible, would be great.

Discussion and Conclusions: The end of this section could benefit from a more accentuated take home message that goes beyond the presentation of the scientific findings. What did we learn from using this high-resolution earth system model compared to the low-resolution ESM2M or regional setups (e.g. Jacox et al. 2015,2016, Frischknecht et al. 2015,2017). I think the paper discussion could benefit from adding a comment to the implications the authors already state.

**Figures**

Generally, the figures all have very cryptic titles and headings (e.g. GFDL ESM2.6 FMA SST (shaded) SLP (contour) Hipass Seas StdAnom, Figure 4). I think cleaning up the figure titles/headings and including the necessary information in the figure caption (while explaining used acronyms and abbreviations) would ease the reader's understanding and greatly help to focus on the relevant things.

Figure 5: In the caption, it says "(see gray boxes in Fig. 6)". These boxes are however not visible in Fig. 6, right? I assume this comment needs to be removed.

Figure 5: The notation of (0/1) reflecting the evolution of an event before and after its peak is hard to grasp in the beginning. Make sure to better introduce this notation or

change the x labels in the figures.

Figure A1: What do the dashed lines in the plots to the right represent?

---

## Referee Comment (RC2) · Anonymous Referee #2 · 11 Oct 2017

**General comments**

Turi et al. use a state-of-the-art Earth System Model to address the effects of ENSO in O2 and pH in coastal waters of the California Current System. They find that the mean drivers at surface differ for both O2 and pH: the O2 response extends for several hundreds of km due to temperature-related changes in solubility, while coastal upwelling affects DIC and drives pH changes within 100 km from the coast. Below 100m depth, the responses of O2 and pH seemed coupled; both responded to changes in isopycnal surfaces (e.g., by coastally trapped waves). I found the approach and results very interesting and sound; I also appreciate the focus on two very important variables (O2 and

pH) and their connections (rather than looking at them in an isolated way). I'd recommend this manuscript for publication after my moderate comments are addressed; in particular, I think that including discussion/conclusions on the large variability between events would strengthen the manuscript.

Specific comments

\*The abstract would benefit from introducing early on why we care about the effect of ENSO on coastal O2 and pH (I'd suggest one or two lines at the very start).

\*Section 2.1:

-Pag4 Line6: What is meant by "prototype"? It gives the impression of a model in the early stages of development – but I don't think it is the case. Please consider explaining better or re-wording.

-Line 18-19: I am wondering why WOA05 was used instead one of the more recent versions (2009, 2013). Also, I suggest referring to WOA data as "climatologies" rather than as "modeled" data.

\*Section 2.2:

-Pag4 Line 23: Why is wintertime NDJ instead of the more common (and likely more winter-appropriate) DJF? (or even JFM).

-Line 24: This sentence makes one wonder: What are the drift issues in the carbonate chemistry? It would be useful to read a line or two about this, to keep the reader from wondering if there is anything wrong with the model and better justify the need of a filter. If the drift happens only at the beginning of the simulation as stated, why not consider those years as "spinup time" and remove them from the analysis? Please explain more or re-word the sentence.

\*Section 2.3:

-P5 Line 10: Please cite source of the climatological SST and SLP

OSD
-Line 19-21: The location of the sections needs to be described.

-Lines 24-29: In line 24, please remove "very" – one could argue that EMS2.6 and ROMS are not "very" similar. Also, while the qualitative description is useful, it would be great to also see a more quantitative comparison (eg, compare mean and ranges of warm-cold ENSO signal for the 3 models)

-Line 30: as asked for wintertime: why is springtime defined as FMA?

-Same line: mention source of CHL observations (SeaWiFS)

-P6 L3-5: I think the lag between ESM2.6 and observations should be mentioned. Furthermore, it would be useful to see the ROMS reanalysis in fig A1 (ideally, it would be closer to the observations and strengthen the justification of its use to evaluate both GFDL's models). The latter is just a suggestion.

\*Section 3, Results:

-P7 L19: Please add a sentence or two to justify the assumption of linearity

-L31-34, From "This difference...": This belongs to the discussion and should be removed from the Results section. It is actually proved later.

-P8 L9: I recommend to rewrite "The pH (Fig 7c, f, i, I) and O2 responses are..."

-L 14: "we next split ... El Niño composite means into their individual components". At first, I thought this meant that you were going to divide again in the individual ENSO events. Please consider re-wording (eg, "into their different drivers")

-L23: I recommend citing page or table from Sarmiento and Gruber (2006), to enable the reader to find the equations easily. Otherwise, add the equations here or in an appendix. Also, while T has a dominant role in solubility, salinity also affects solubility and I suggest to mention it (it would be great if the S role could be quantified as well!).

-L32-32: The residual effect: does it also consider the effect of winds?

OSD
-P9 L12: Comment only – it's unfortunate that DIC and Alk couldn't be saved beyond the surface. If these simulations are run again in the future, maybe those two variables could be saved instead of [H+] (if disk space allows for one extra output).

-L15-16: The partial derivatives of pH are key and deserve a more explicit description of how they were calculated. The cited CO2calc is just a calculator of the carbonate system. How did you alter DIC/Alk/T/S in order to calculate the changes on pH?

-L27-29: does this call for a reference to Fig 6a?

\*Section 4, Discussion and Conclusions:

-Most of the analysis was performed for the mean ENSO signal (ie, composite of all ENSO events), so the conclusions are mostly based on this mean. However, the manuscript also describes early on large differences between events (Section 3.1). It would be beneficial for the manuscript to expand the conclusions in terms of this large variability between events (e.g., are the processes identified as responsible for the mean signal be still dominant in all the individual events? Any suggestions on the causes of the variability?)

-Note that in the Results section there is a lot of discussion. You could either remove the discussion parts in the Results (as suggested above for a particular case, but there are more instances), or you could also move all discussions to Section 3 and rename it "Results and Discussion". In the latter case, Section 4 would be a shorter Conclusions section.

-P10 L29-30: Could your differences with respect to Nam et al (2011) be based on the fact that you work with a mean ENSO signal and they focus on a specific event?

Technical comments

\*I'd suggest to rewrite "100 km" as "hundreds of kilometers" where "100" intends to mean "hundreds" rather than "one hundred" (e.g., in the abstract "reaches up to several 100 km offshore"; also in section 4)

OSD
\*Pag4 Line 18: I think it'd be better to refer to "the beginning of year 141 of \*a\* CM2.6 1900 control simulation", rather than \*the\*, because "\*the\* control simulation" makes me think of the control run being described (ESM2.6 control simulation)

\*Pag4 Lines 30-32: I suggest re-ordering this sentence: first state the need to interpret patterns of ENSO-related signals, and then explain that to this end, you use standard-ized anomalies.

\*P5 Lines 5-6: By now, the models have been introduced and are referred to as EMS2.6 and ESM2M. Please remove the "GFDL-"; same for figure captions.

\*P5 L31: "does not" instead of "doesn't"

\*P8 L10: should it be 32 and 36 degrees N?

\*P9 L26: replace "seems to mainly act" by "mainly acts"

\*Figures

-Remove titles in the figures; make sure captions capture all the information in the current titles. In Fig 2, make a legend for the density contours (if the latter is not possible, then keep only the "Density (contours)..." text in the title).

-Fig 5: Explain in the caption the zeros and ones found in the x labels. The caption says "gray" outline in g, but it looks like black. Also, note that the solid line is hard to distinguish from the dashed ones on the screen (it's ok in the printed version) -1 suggest to make solid lines thicker if possible.

-Why there are two figures labeled A1 and A2, if there is no Appendix section? Shouldn't these figures be labeled consecutively, following the order in which they were mentioned?

\*References: I did not check the references thoroughly, but spotted a mistake in Robbins et al (2010): it says "Max" instead of "Mac" OSD

---

## Referee Comment (RC3) · Anonymous Referee #3 · 14 Oct 2017

1. General Comments: Turi and colleagues conduct here a well-detailed and interesting analysis of how ENSO impacts the temperature, O2, and pH field structures of the California Current System (CCS). The paper's focus on temperature, O2 and pH and driving mechanisms is highly relevant to attribution and descriptive studies of the CCS, given ecosystems' vulnerability to changes in these variables, and thus should generate a broad and interested audience. Specifically, Turi et al. reveal significant model improvement in representing ENSO physical variability of the CCS in a coupled high-resolution model (vs. CMIP5-type resolution), which they use to evaluate the diversity and mechanisms driving ENSO impacts off the California coast. The authors uncover large variations in the CCS response, a point that is somewhat under-developed in

the paper and should be further elaborated on given its high relevance to CCS-ENSO studies. Using a composite analysis of the 3-D spatial structure and component decomposition of O2 and pH anomalies from their simulated ENSO events, they suggest different mechanisms driving O2 and pH anomalies at different depths, with changes in temperature as a major driver of surface O2 anomalies, while changes in isopycnals depth and upwelling accounting for most of the variability in pH and O2 at depth. Overall, the paper by Turi and colleagues is well written, the approach is novel, and results are thought provoking, though I felt the discussion section could be further developed given their interesting results and their relevance to other CCS studies. This paper is suitable for publication in the Journal of Ocean Science and I recommend strengthening it with the following comments, revision, and suggestions below.

2. Specific Comments:

1) The paper is appropriately and well titled, but since temperature is so prevalently used in figures and discussed throughout the paper, and since temperature is also an important ecosystem stressor, perhaps it ought to be in the title as well?

2) The introduction provides a thorough review of previous work, and could perhaps be improved by adding a few lines on processes driving O2 and pH variability in the upper 150 m of the CCS (i.e. upwelling, solubility, and productivity and respiration, etc.). This would help putting the processes section in context.

3) The method section could use more detailed description of the model and its configuration, e.g. : what is the model's vertical resolution? How long as the model been spun up for? what is the general structure of the BGC model?

4) It would also be helpful to explain the choice of using a coupled configuration vs. a hindcast simulation (CORE2/NCEP-forced run) of the high resolution model. Wouldn't a hindcast run provide a more realistic representation of ENSO impacts on ocean biogeochemistry and physics? This would also allow for more appropriate comparison to observations.

5) The authors extensively uses FMA anomalies without justifying the choice of this season/ period. Is this associated with the time scales (2-3 months) of coastal wave propagation from the equatorial region post the maximum equatorial SST anomaly typically observed in DJF? Or is this simply based on the timing of the maximum CCS impact as shown in the mean response in Fig 5? This is especially confusing as some variables are plotted in FMA (SST, O2, pH) while others are shown for DJF (e.g. SLP). FMA is also described as spring, but spring is typically MAM, and winter is DJF. Please explicitly state the choice for FMA, and describe acronyms somewhere in paper/figures (FMA=February-March-April, etc.).

6) Fig A1 ought to be within the paper rather than a supplementary or appendix since this seems to be a major deficiency in the model and should be made more visible and relevant. Additionally, the method section could also benefit from a comparison of simulated BGC fields to the WOA climatologies, i.e. how large are the BGC biases, and how do they differ in the high resolution vs. low resolution version of the GFDL model, at least for the CCS. A discussion of the implications of model biases on the paper's results could help provide a more thorough overview of the potential and limitations of the authors' approach, especially when relating their results to observations.

7) The diversity of the ENSO SST and SLP anomalies shown in Figure 4 is very interesting, and so is the diversity of the averaged O2 and pH changes shown in Fig 5. It would be useful and highly relevant to see similar maps as shown for SST and SLP (as Fig 4) for O2 and pH for different events (perhaps in Appendix, but preferably in the paper). This is perhaps most useful to inform observations-based studies which are often limited to few or single ENSO events. Generally, the diverse response in BGC should be detailed further and reasons for this diversity could also be explored, especially since this is one of the paper's main stated and novel research questions. e.g. What were the initial conditions prior to each event? Do similar patterns emerge in the CCS from different ENSO events (eastern vs. central El Niño)? Do both O2 and pH show the same degree of variability as SST and SLP? The diversity of SST and SLP

to ENSO events could also be shown for the observations, and would be interesting to assess whether such high variations across ENSO events differs in obs. vs model.

8) At the same time, the diversity of the CCS response to ENSO questions the use of the composite mean difference to evaluate "typical" ENSO impacts; i.e. how representative is the composite mean of the ENSO anomalies used in Fig 6-10. Perhaps adding a statistical test/stipplings to show which of these patterns are significant could help address this?

9) In page 9 line 5-6, the authors propose deepening of the thermocline during El Niño to explain the increase in O2 at 100m all along the coast, but for pH changes, they invoke a dipole in upwelling north vs. south 40oN (Pg9 L 25). This is confusing since changes in intensity or source of upwelling and isopycnal depths should impact pH and O2 similarly. How do the authors reconcile this discrepancy?

10) The process analysis conducted here is valuable in understanding the CCS biogeochemical response to ENSO physical changes. Important questions on which of these physical processes drive these biogeochemical anomalies however remain unclear, and perhaps could be discussed further. e.g., what is the role of "remote" wave propagation vs "local" atmospheric forcing of upwelling on the biogeochemical anomalies presented here? This could be addressed using existing figures or editing figures, e.g. superimposing SLP anomalies on BGC anomalies to assess role of atmospheric forcing effects on spatial anomalies in pH and O2. The analysis of Frischknecht et al (2015) regarding the roles of remote vs local forcing in driving physical and biogeochemical anomalies could also be discussed in relation to Turi et al's regionally distinct imprints of ENSO on CA CCS.

11) Another important question that belong to the mechanisms section and discussion but is unclear is what is the role of changes in transport vs. changes in biological production and respiration rates on O2 anomalies? In an MITgcm hindcast simulation, Ito and Deutsch (2013) decompose O2 changes due to ENSO to changes in respiration rates, transport, and solubility in the northern tropical Pacific OMZ and show that a warmer thermocline is also more oxygenated, in agreement with Turi et al's model results. They argue however that during El Niño, declines in O2 respiration rate in the thermocline associated with reduced carbon export that result from a deeper thermocline, reduced nutrients export to surface and reduced productivity, is the main driver of O2 changes. The heaving of isopycnal shown and suggested by Turi goes in the same direction but doesn't preclude reinforcing biological effects from being a contributing or dominant component.

12) Generally, the discussion section could benefit from expanding on how these results fit in the context of other studies' findings. The diversity of ENSO events is especially relevant to past and future studies of ENSO and the CCS, mainly that a generic CCS response to ENSO shouldn't be expected given effects of initial local conditions, different teleconnections, etc.

13) The figure titles and captions are hard to read for quick readers, and could really use more attention to explaining acronyms, reducing repetitions, and clarifying what the figure is trying to convey. e.g. the terms "high-pass filtered standardized" is already stated in methods and needs not be repeated in each figure.

3. Technical Corrections: 1) Pg 4 Line 9, what is vertical resolution?

2) Pg 4 Line 18: Do authors mean "observed climatologies of O2, nitrate, etc."? To my knowledge, WOA doesn't include modeled fields.

3) Fig 2. Caption Line 2: "ROMS Climatology"? Shouldn't it be an anomaly rather than a climatology?

4) Fig 5, "gray box", do authors mean Fig 5g?

5) Pg 6. L20." magnitude of +/- sigma". Sigma from area average?

6) Page 7 "Fig 5b and e" or "5b" only?

7) Figure 3 and chlorophyll seems less relevant to the paper's theme and could be delegated to Appendix/supplementary.

---

## Author Comment (AC1) · 12 Dec 2017

**Anonymous Reviewer #1 Received and published: 30 August 2017**

The reviewer's comments are in *italics* and the Authors' replies are in plain text.

The authors would like to thank anonymous reviewer #1 for her/his thorough review of the manuscript and for the insightful comments that have greatly helped improve the publication.

**1. Summary**

Turi et al. use a high-resolution fully-coupled global earth system model (GFDLESM2.6) to uncover the O2 and pH response of the California Current System (CalCS) to El Niño/Southern Oscillation (ENSO). Despite significant variations of the response to individual events due to ENSO diversity, composite means for warm/cold events reveal consistent physical and biogeochemical changes along the US West Coast. While the O2 response to ENSO is wide-spread and differs between the surface (driven by changes in the solubility) and at 100m (driven by changes in the thermocline structure), the pH response is mainly confined to the coastal environment, highlighting the dominant role of changes in dissolved inorganic carbon (DIC) and upwelling associated with ENSO.

**2. General comments**

The influence of ENSO on the physical and biogeochemical environment of the CalCS is evidenced both in observational and model-based data. Investigating the associated changes in the current system improves our understanding of the ecosystem functioning, its sensitivity to change and is relevant for ocean management. Even though the imprint of ENSO on the CalCS has been thoroughly studied, the diversity among ENSO events and the lack of high-frequency 3-dimensional observational records render modeling studies (such as the one presented) extremely valuable in providing opportunities to study ecosystem changes on interannual time scales. The paper presents results on O2 and pH changes in the CalCS based on a global high-resolution fully coupled earth system model, which is, to my knowledge, unprecedented. In my opinion, it thus addresses relevant scientific questions within the scope of Ocean Science. The work is generally well presented and structured. I recommend this manuscript for publication with Ocean Science after addressing the few minor comments on the manuscript listed below.

**3. Specific and purely technical comments**

**Methods**

Page 5, Line 1: What do you mean by "interannual standard deviations"? I assume you apply the Lanczos filter as described, deseasonalize the data which yields time series that retain only anomalies on interannual time scales, right? From these time series, you compute standard deviations and use them to normalize the response? I guess the term "interannual standard deviations" got me confused and you might want to clarify this passage, as the normalization of all the data is key to interpreting your results.

The standardized anomalies during February-March-April (FMA) are computed starting with monthly data. A monthly mean climatology is subtracted for each month creating a monthly mean time series of anomalies. The Lanczos time filtering is now applied, which removes variability on time scales longer than 10 years. FMA averages of time filtered anomalies are created for each year. The inter-annual standard deviation of the FMA anomalies is computed and used to standardize (normalize) the FMA time-filtered anomalies.

We also removed the clause "as they were on the lower end of the frequency spectrum" from this paragraph, which could have made the description less clear.

**Model Evaluation**

The authors demonstrate that ESM2.6 shows an improved coastal ENSO response compared to ESM2M. They also compare their model results to a data-assimilative ROMS hindcast. Can the authors elaborate on the reason of using the data assimilative regional model for means of comparison? Would e.g. the long (though not as high-frequent) CalCOFI records provide an additional opportunity for a model independent evaluation of the presented response?

Our main reason for using the ROMS reanalysis was that it is the best available spatio-temporally resolved estimate of the physical ocean state off the US West Coast. Individual data sources provide reasonably well resolved information on portions of the water column (e.g., satellite data for the surface), or poorly resolved information on the full water column (e.g., CalCOFI, Argo, etc.). The ROMS reanalysis assimilates these data (including CalCOFI), so it combines the strengths of the observations with the strengths of an unconstrained (no data assimilation) ocean model, to give a product that is more useful than either of those alone.

Page 6, Lines 5-10: Nino 3.4 variance seems substantially overestimated in the global models. Moreover, ENSO seems to be more periodic with a maximum in the frequency spectrum at 3 years return time (Figure A1). I would like to encourage the authors to elaborate on this fact and the potential implications of this particular issue in the model for their interpretation of the presented results in more detail.

Yes, the ENSO variance is overestimated and events are more periodic in both versions of the models, especially GFDL ESM2.6, compared to observations. These aspects of the model are discussed in the text and shown in the supplemental material. First, based on long model runs it is not clear that one can fully describe ENSO's spectra with ~50 years of data (Wittenberg 2009, also see the spread in the spectra from 52-year segments of the 500-year ESM2M in Fig. S1). While it is still

likely that the SST variance in Nino 3.4 is too large, the atmospheric response in the sea level pressure over the North Pacific is not as large relative to observations. In addition, we are not able to compare the extratropical coastally trapped ocean wave response to observations. So, the we are not able to fully evaluate the influence of the amplitude of ENSO events on the CCS. However, to address this issue we have normalized the fields examined here by the local standard deviation, which should partly compensate for differences in magnitudes of events between the model simulations and observations.

A more quantitative model comparison between GFDL-ESM2.6, GFDL-ESM2M, and the ROMS reanalysis goes beyond the scope and focus of this manuscript, and is anticipated to be the focus of future publications on ESM2.6.

Figure 5: Plot a) showing SSTs could basically be backed up by satellite observations, not? I was surprised to see such a large spread among different events, in particular in the phasing of the peak anomalies (is this ESM specific, or does it really reflect the ENSO diversity?). It would be interesting to see whether observations show similar differences, or whether the phasing is more synchronous.

The spread is indeed large, and is also seen in observations, as can be seen in the following Figure showing standardized anomalies of Hadley SST anomalies (a combination of satellite and in situ data) for the top nine El Niño events. The difference in circulation can result from a number of factors including ENSO diversity, differences in SST anomalies in the tropical Pacific and random fluctuations in the extratropical atmosphere and ocean ("climate noise"). This aspect of the differences between events is now discussed in greater detail in the last paragraph in the paper.